# Learning Multiple Tasks with Multilinear Relationship Networks

**Mingsheng Long, Zhangjie Cao, Jianmin Wang, Philip S. Yu**
School of Software, Tsinghua University, Beijing 100084, China
{mingsheng,jimwang}@tsinghua.edu.cn  caozhangjie14@gmail.com  psyu@uic.edu

## Abstract

Deep networks trained on large-scale data can learn transferable features to promote learning multiple tasks. Since deep features eventually transition from general to specific along deep networks, a fundamental problem of multi-task learning is how to exploit the task relatedness underlying parameter tensors and improve feature transferability in the multiple task-specific layers. This paper presents Multilinear Relationship Networks (MRN) that discover the task relationships based on novel tensor normal priors over parameter tensors of multiple task-specific layers in deep convolutional networks. By jointly learning transferable features and multilinear relationships of tasks and features, MRN is able to alleviate the dilemma of negative-transfer in the feature layers and under-transfer in the classifier layer. Experiments show that MRN yields state-of-the-art results on three multi-task learning datasets.

## 1 Introduction

Supervised learning machines trained with limited labeled samples are prone to overfitting, while manual labeling of sufficient training data for new domains is often prohibitive. Thus it is imperative to design versatile algorithms for reducing the labeling consumption, typically by leveraging off-the-shelf labeled data from relevant tasks. Multi-task learning is based on the idea that the performance of one task can be improved using related tasks as inductive bias [4]. Knowing the task relationship should enable the transfer of shared knowledge from relevant tasks such that only task-specific features need to be learned. This fundamental idea of task relatedness has motivated a variety of methods, including multi-task feature learning that learns a shared feature representation [1, 2, 6, 5, 23], and multi-task relationship learning that models inherent task relationship [10, 14, 29, 31, 15, 17, 8].

Learning inherent task relatedness is a hard problem, since the training data of different tasks may be sampled from different distributions and fitted by different models. Without prior knowledge on the task relatedness, the distribution shift may pose a major difficulty in transferring knowledge across different tasks. Unfortunately, if cross-task knowledge transfer is impossible, then we will overfit each task due to limited amount of labeled data. One way to circumvent this dilemma is to use an external data source, e.g. ImageNet, to learn transferable features through which the shift in the inductive biases can be reduced such that different tasks can be correlated more effectively. This idea has motivated some latest deep learning methods for learning multiple tasks [25, 22, 7, 27], which learn a shared representation in feature layers and multiple independent classifiers in classifier layer.

However, these deep multi-task learning methods do not explicitly model the task relationships. This may result in under-transfer in the classifier layer as knowledge can not be transferred across different classifiers. Recent research also reveals that deep features eventually transition from general to specific along the network, and feature transferability drops significantly in higher layers with increasing task dissimilarity [28], hence the sharing of all feature layers may be risky to negative-transfer. Therefore, it remains an open problem how to exploit the task relationship across different deep networks while improving the feature transferability in task-specific layers of the deep networks.

This paper presents Multilinear Relationship Network (MRN) for multi-task learning, which discovers the task relationships based on multiple task-specific layers of deep convolutional neural networks. Since the parameters of deep networks are natively tensors, the tensor normal distribution [21] is explored for multi-task learning, which is imposed as the prior distribution over network parameters of all task-specific layers to learn find-grained multilinear relationships of tasks, classes and features. By jointly learning transferable features and multilinear relationships, MRN is able to circumvent the dilemma of negative-transfer in feature layers and under-transfer in classifier layer. Experiments show that MRN learns fine-grained relationships and yields state-of-the-art results on standard benchmarks.

## 2   Related Work

Multi-task learning is a learning paradigm that learns multiple tasks jointly by exploiting the shared structures to improve generalization performance [4, 19] and mitigate manual labeling consumption. There are generally two categories of approaches: **(1)** multi-task feature learning, which learns a shared feature representation such that the distribution shift across different tasks can be reduced [1, 2, 6, 5, 23]; **(2)** multi-task relationship learning, which explicitly models the task relationship in the forms of task grouping [14, 15, 17] or task covariance [10, 29, 31, 8]. While these methods have achieved improved performance, they may be restricted by their shallow learning paradigm that cannot embody task relationships by suppressing the task-specific variations in transferable features.

Deep networks learn abstract representations that disentangle and hide explanatory factors of variation behind data [3, 16]. Deep representations manifest invariant factors underlying different populations and are transferable across similar tasks [28]. Thus deep networks have been successfully explored for domain adaptation [11, 18] and multi-task learning [25, 22, 32, 7, 20, 27], where significant performance gains have been witnessed. Most multi-task deep learning methods [22, 32, 7] learn a shared representation in the feature layers and multiple independent classifiers in the classifier layer without inferring the task relationships. However, this may result in *under-transfer* in the classifier layer as knowledge cannot be adaptively propagated across different classifiers, while the sharing of all feature layers may still be vulnerable to *negative-transfer* in the feature layers, as the higher layers of deep networks are tailored to fit task-specific structures and may not be safely transferable [28].

This paper presents a multilinear relationship network based on novel tensor normal priors to learn transferable features and task relationships that mitigate both under-transfer and negative-transfer. Our work contrasts from prior relationship learning [29, 31] and multi-task deep learning [22, 32, 7, 27] methods in two key aspects. **(1)** Tensor normal prior: our work is the first to explore tensor normal distribution as priors of network parameters in different layers to learn multilinear task relationships in deep networks. Since the network parameters of multiple tasks natively stack into high-order tensors, previous matrix normal distribution [13] cannot be used as priors of network parameters to learn task relationships. **(2)** Deep task relationship: we define the tensor normal prior on multiple task-specific layers, while previous deep learning methods do not learn the task relationships. To our knowledge, multi-task deep learning by tensor factorization [27] is the first work that tackles multi-task deep learning by tensor factorization, which learns shared feature subspace from multilayer parameter tensors; in contrast, our work learns multilinear task relationships from multiplayer parameter tensors.

## 3   Tensor Normal Distribution

### 3.1   Probability Density Function

Tensor normal distribution is a natural extension of multivariate normal distribution and matrix-variate normal distribution [13] to tensor-variate distributions. The multivariate normal distribution is order-1 tensor normal distribution, and matrix-variate normal distribution is order-2 tensor normal distribution. Before defining tensor normal distribution, we first introduce the notations and operations of order-$K$ tensor. An order-$K$ tensor is an element of the tensor product of $K$ vector spaces, each of which has its own coordinate system. A vector $\mathbf{x} \in \mathbb{R}^{d_1}$ is an order-1 tensor with dimension $d_1$. A matrix $\mathbf{X} \in \mathbb{R}^{d_1 \times d_2}$ is an order-2 tensor with dimensions $(d_1, d_2)$. A order-$K$ tensor $\mathcal{X} \in \mathbb{R}^{d_1 \times \ldots \times d_K}$ with dimensions $(d_1, \ldots, d_K)$ has elements $\{x_{i_1 \ldots i_K} : i_k = 1, \ldots, d_k\}$. The vectorization of $\mathcal{X}$ is unfolding the tensor into a vector, denoted by $\text{vec}(\mathcal{X})$. The matricization of $\mathcal{X}$ is a generalization of vectorization, reordering the elements of $\mathcal{X}$ into a matrix. In this paper, to simply the notations and

describe the tensor relationships, we use the mode-$k$ matricization and denote by $\mathbf{X}_{(k)}$ the mode-$k$ matrix of tensor $\mathcal{X}$, where row $i$ of $\mathbf{X}_{(k)}$ contains all elements of $\mathcal{X}$ having the $k$-th index equal to $i$.

Consider an order-$K$ tensor $\mathcal{X} \in \mathbb{R}^{d_1 \times \ldots \times d_K}$. Since we can vectorize $\mathcal{X}$ to a $(\prod_{k=1}^{K} d_k) \times 1$ vector, the normal distribution on a tensor $\mathcal{X}$ can be considered as a multivariate normal distribution on vector $\text{vec}(\mathcal{X})$ of dimension $\prod_{k=1}^{K} d_k$. However, such an ordinary multivariate normal distribution ignores the special structure of $\mathcal{X}$ as a $d_1 \times \ldots \times d_K$ tensor, and as a result, the covariance characterizing the correlations across elements of $\mathcal{X}$ is of size $(\prod_{k=1}^{K} d_k) \times (\prod_{k=1}^{K} d_k)$, which is often prohibitively large for modeling and estimation. To exploit the structure of $\mathcal{X}$, tensor normal distributions assume that the $(\prod_{k=1}^{K} d_k) \times (\prod_{k=1}^{K} d_k)$ covariance matrix $\mathbf{\Sigma}_{1:K}$ can be decomposed into the Kronecker product $\mathbf{\Sigma}_{1:K} = \mathbf{\Sigma}_1 \otimes \ldots \otimes \mathbf{\Sigma}_K$, and elements of $\mathcal{X}$ (in vectorization) follow the normal distribution,

$$\text{vec}\left(\mathcal{X}\right) \sim \mathcal{N}\left(\text{vec}\left(\mathcal{M}\right), \mathbf{\Sigma}_1 \otimes \ldots \otimes \mathbf{\Sigma}_K\right), \tag{1}$$

where $\otimes$ is the Kronecker product, $\mathbf{\Sigma}_k \in \mathbb{R}^{d_k \times d_k}$ is a positive definite matrix indicating the covariance between the $d_k$ rows of the mode-$k$ matricization $\mathbf{X}_{(k)}$ of dimension $d_k \times (\prod_{k' \neq k} d_{k'})$, and $\mathcal{M}$ is a mean tensor containing the expectation of each element of $\mathcal{X}$. Due to the decomposition of covariance as the Kronecker product, the tensor normal distribution of an order-$K$ tensor $\mathcal{X}$, parameterized by mean tensor $\mathcal{M}$ and covariance matrices $\mathbf{\Sigma}_1, \ldots, \mathbf{\Sigma}_K$, can define probability density function as [21]

$$p\left(\mathbf{x}\right) = (2\pi)^{-d/2} \left(\prod_{k=1}^{K} |\mathbf{\Sigma}_k|^{-d/(2d_k)}\right) \times \exp\left(-\frac{1}{2}(\mathbf{x} - \boldsymbol{\mu})^{\mathsf{T}} \mathbf{\Sigma}_{1:K}^{-1}(\mathbf{x} - \boldsymbol{\mu})\right), \tag{2}$$

where $|\cdot|$ is the determinant of a square matrix, and $\mathbf{x} = \text{vec}\left(\mathcal{X}\right), \boldsymbol{\mu} = \text{vec}\left(\mathcal{M}\right), \mathbf{\Sigma}_{1:K} = \mathbf{\Sigma}_1 \otimes \ldots \otimes \mathbf{\Sigma}_K, d = \prod_{k=1}^{K} d_k$. The tensor normal distribution corresponds to the multivariate normal distribution with Kronecker decomposable covariance structure. $\mathcal{X}$ following tensor normal distribution, i.e. $\text{vec}\left(\mathcal{X}\right)$ following the normal distribution with Kronecker decomposable covariance, is denoted by

$$\mathcal{X} \sim \mathcal{TN}_{d_1 \times \ldots \times d_K}\left(\mathcal{M}, \mathbf{\Sigma}_1, \ldots, \mathbf{\Sigma}_K\right). \tag{3}$$

## 3.2 Maximum Likelihood Estimation

Consider a set of $n$ samples $\{\mathcal{X}_i\}_{i=1}^{n}$ where each $\mathcal{X}_i$ is an order-3 tensor generated by a tensor normal distribution as in Equation (2). The maximum likelihood estimation (MLE) of the mean tensor $\mathcal{M}$ is

$$\widehat{\mathcal{M}} = \frac{1}{n} \sum_{i=1}^{n} \mathcal{X}_i. \tag{4}$$

The MLE of covariance matrices $\widehat{\mathbf{\Sigma}}_1, \ldots, \widehat{\mathbf{\Sigma}}_3$ are computed by iteratively updating these equations:

$$\widehat{\mathbf{\Sigma}}_1 = \frac{1}{nd_2d_3} \sum_{i=1}^{n} (\mathcal{X}_i - \mathcal{M})_{(1)} \left(\widehat{\mathbf{\Sigma}}_3 \otimes \widehat{\mathbf{\Sigma}}_2\right)^{-1} (\mathcal{X}_i - \mathcal{M})_{(1)}^{\mathsf{T}},$$

$$\widehat{\mathbf{\Sigma}}_2 = \frac{1}{nd_1d_3} \sum_{i=1}^{n} (\mathcal{X}_i - \mathcal{M})_{(2)} \left(\widehat{\mathbf{\Sigma}}_3 \otimes \widehat{\mathbf{\Sigma}}_1\right)^{-1} (\mathcal{X}_i - \mathcal{M})_{(2)}^{\mathsf{T}}, \tag{5}$$

$$\widehat{\mathbf{\Sigma}}_3 = \frac{1}{nd_1d_2} \sum_{i=1}^{n} (\mathcal{X}_i - \mathcal{M})_{(3)} \left(\widehat{\mathbf{\Sigma}}_2 \otimes \widehat{\mathbf{\Sigma}}_1\right)^{-1} (\mathcal{X}_i - \mathcal{M})_{(3)}^{\mathsf{T}}.$$

This flip-flop algorithm [21] is efficient to solve by simple matrix manipulations and convergence is guaranteed. Covariance matrices $\widehat{\mathbf{\Sigma}}_1, \ldots, \widehat{\mathbf{\Sigma}}_3$ are not identifiable and the solutions to maximizing density function (2) are not unique, while only the Kronecker product $\mathbf{\Sigma}_1 \otimes \ldots \otimes \mathbf{\Sigma}_K$ (1) is identifiable.

## 4 Multilinear Relationship Networks

This work models multiple tasks by jointly learning transferable representations and task relationships. Given $T$ tasks with training data $\{\mathcal{X}_t, \mathcal{Y}_t\}_{t=1}^{T}$, where $\mathcal{X}_t = \{\mathbf{x}_1^t, \ldots, \mathbf{x}_{N_t}^t\}$ and $\mathcal{Y}_t = \{\mathbf{y}_1^t, \ldots, \mathbf{y}_{N_t}^t\}$ are the $N_t$ training examples and associated labels of the $t$-th task, respectively drawn from $D$-dimensional feature space and $C$-cardinality label space, i.e. each training example $\mathbf{x}_n^t \in \mathbb{R}^D$ and $\mathbf{y}_n^t \in \{1, \ldots, C\}$. Our goal is to build a deep network for multiple tasks $\mathbf{y}_n^t = f_t(\mathbf{x}_n^t)$ which learns transferable features and adaptive task relationships to bridge different tasks effectively and robustly.

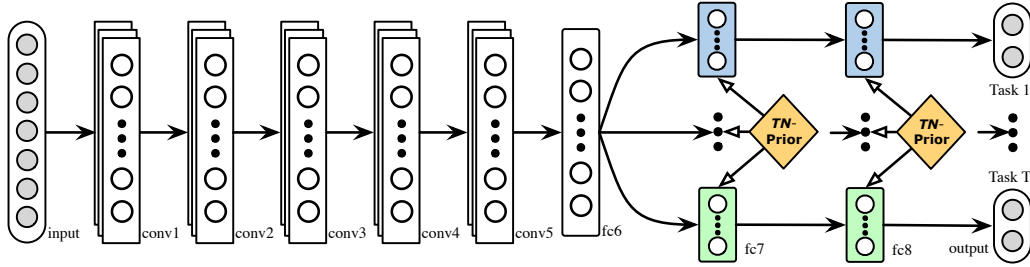

Figure 1: Multilinear relationship network (MRN) for multi-task learning: (1) convolutional layers $conv1$–$conv5$ and fully-connected layer $fc6$ learn transferable features, so their parameters are shared across tasks; (2) fully-connected layers $fc7$–$fc8$ fit task-specific structures, so their parameters are modeled by tensor normal priors for learning multilinear relationships of features, classes and tasks.

## 4.1 Model

We start with deep convolutional neural networks (CNNs) [16], a family of models to learn transferable features that are well adaptive to multiple tasks [32, 28, 18, 27]. The main challenge is that in multi-task learning, each task is provided with a limited amount of labeled data, which is insufficient to build reliable classifiers without overfitting. In this sense, it is vital to model the task relationships through which each pair of tasks can help with each other to enable knowledge transfer if they are related, and can remain independent to mitigate negative transfer if they are unrelated. With this idea, we design a Multilinear Relationship Network (MRN) that exploits both feature transferability and task relationship to establish effective and robust multi-task learning. Figure 1 shows the architecture of the proposed MRN model based on AlexNet [16], while other deep networks are also applicable.

We build the proposed MRN model upon AlexNet [16], which is comprised of convolutional layers ($conv1$–$conv5$) and fully-connected layers ($fc6$–$fc8$). The $\ell$-th $fc$ layer learns a nonlinear mapping $\mathbf{h}_n^{t,\ell} = a^{\ell}\left(\mathbf{W}^{t,\ell}\mathbf{h}_n^{t,\ell-1} + \mathbf{b}^{t,\ell}\right)$ for task $t$, where $\mathbf{h}_n^{t,\ell}$ is the hidden representation of each point $\mathbf{x}_n^t$, $\mathbf{W}^{t,\ell}$ and $\mathbf{b}^{t,\ell}$ are the weight and bias parameters, and $a^{\ell}$ is the activation function, taken as ReLU $a^{\ell}(\mathbf{x}) = \max(\mathbf{0}, \mathbf{x})$ for hidden layers or softmax units $a^{\ell}(\mathbf{x}) = e^{\mathbf{x}} / \sum_{j=1}^{|\mathbf{x}|} e^{x_j}$ for the output layer. Denote by $\mathbf{y} = f_t(\mathbf{x})$ the CNN classifier of $t$-th task, and the empirical error of CNN on $\{\mathcal{X}_t, \mathcal{Y}_t\}$ is

$$\min_{f_t} \sum_{n=1}^{N_t} J\left(f_t\left(\mathbf{x}_n^t\right), \mathbf{y}_n^t\right), \tag{6}$$

where $J$ is the cross-entropy loss function, and $f_t\left(\mathbf{x}_n^t\right)$ is the conditional probability that CNN assigns $\mathbf{x}_n^t$ to label $\mathbf{y}_n^t$. We will not describe how to compute the convolutional layers since these layers can learn transferable features in general [28, 18], and we will simply share the network parameters of these layers across different tasks, without explicitly modeling the relationships of features and tasks in these layers. To benefit from pre-training and fine-tuning as most deep learning work, we copy these layers from a model pre-trained on ImageNet 2012 [28], and fine-tune all $conv1$–$conv5$ layers.

As revealed by the recent literature findings [28], the deep features in standard CNNs must eventually transition from general to specific along the network, and the feature transferability decreases while the task discrepancy increases, making the features in higher layers $fc7$–$fc8$ unsafely transferable across different tasks. In other words, the $fc$ layers are tailored to their original task at the expense of degraded performance on the target task, which may deteriorate multi-task learning based on deep neural networks. Most previous methods generally assume that the multiple tasks can be well correlated given the shared representation learned by the feature layers $conv1$–$fc7$ of deep networks [25, 22, 32, 27]. However, it may be vulnerable if different tasks are not well correlated under deep features, which is common as higher layers are not safely transferable and tasks may be dissimilar. Moreover, existing multi-task learning methods are natively designed for binary classification tasks, which are not good choices as deep networks mainly adopt multi-class softmax regression. It remains an open problem to explore the task relationships of multi-class classification for multi-task learning.

In this work, we jointly learn transferable features and multilinear relationships of features and tasks for multiple task-specific layers $\mathcal{L}$ in a Bayesian framework. Based on the transferability of deep

networks discussed above, the task-specific layers $\mathcal{L}$ are set to $\{fc7, fc8\}$. Denote by $\mathcal{X} = \{\mathcal{X}_t\}_{t=1}^T$, $\mathcal{Y} = \{\mathcal{Y}_t\}_{t=1}^T$ the complete training data of $T$ tasks, and by $\mathbf{W}^{t,\ell} \in \mathbb{R}^{D_1^\ell \times D_2^\ell}$ the network parameters of the $t$-th task in the $\ell$-th layer, where $D_1^\ell$ and $D_2^\ell$ are the rows and columns of matrix $\mathbf{W}^{t,\ell}$. In order to capture the task relationship in the network parameters of all $T$ tasks, we construct the $\ell$-th layer parameter *tensor* as $\mathcal{W}^\ell = \left[\mathbf{W}^{1,\ell}; \ldots; \mathbf{W}^{T,\ell}\right] \in \mathbb{R}^{D_1^\ell \times D_2^\ell \times T}$. Denote by $\mathcal{W} = \{\mathcal{W}^\ell : \ell \in \mathcal{L}\}$ the set of parameter tensors of all the task-specific layers $\mathcal{L} = \{fc7, fc8\}$. The Maximum a Posteriori (MAP) estimation of network parameters $\mathcal{W}$ given training data $\{\mathcal{X}, \mathcal{Y}\}$ for learning multiple tasks is

$$
\begin{aligned}
p\left(\mathcal{W} | \mathcal{X}, \mathcal{Y}\right) &\propto p\left(\mathcal{W}\right) \cdot p\left(\mathcal{Y} | \mathcal{X}, \mathcal{W}\right) \\
&= \prod_{\ell \in \mathcal{L}} p\left(\mathcal{W}^\ell\right) \cdot \prod_{t=1}^T \prod_{n=1}^{N_t} p\left(\mathbf{y}_n^t | \mathbf{x}_n^t, \mathcal{W}^\ell\right),
\end{aligned}
\tag{7}
$$

where we assume that for prior $p\left(\mathcal{W}\right)$, the parameter tensor of each layer $\mathcal{W}^\ell$ is independent on the parameter tensors of the other layers $\mathcal{W}^{\ell' \neq \ell}$, which is a common assumption made by most feed-forward neural network methods [3]. Finally, we assume when the network parameter is sampled from the prior, all tasks are independent. These independence assumptions lead to the factorization of the posteriori in Equation (7), which make the final MAP estimation in deep networks easy to solve.

The maximum likelihood estimation (MLE) part $p\left(\mathcal{Y} | \mathcal{X}, \mathcal{W}\right)$ in Equation (7) is modeled by deep CNN in Equation (6), which can learn transferable features in lower layers for multi-task learning. We opt to share the network parameters of all these layers ($conv1$–$fc6$). This parameter sharing strategy is a relaxation of existing deep multi-task learning methods [22, 32, 7], which share all the feature layers except for the classifier layer. We do not share task-specific layers (the last feature layer $fc7$ and classifier layer $fc8$), with the expectation to potentially mitigate negative-transfer [28].

The prior part $p\left(\mathcal{W}\right)$ in Equation (7) is the key to enabling multi-task deep learning since this prior part should be able to model the multilinear relationship across parameter tensors. This paper, for the first time, defines the prior for the $\ell$-th layer parameter tensor by *tensor normal distribution* [21] as

$$
p\left(\mathcal{W}^\ell\right) = \mathcal{TN}_{D_1^\ell \times D_2^\ell \times T}\left(\mathbf{O}, \boldsymbol{\Sigma}_1^\ell, \boldsymbol{\Sigma}_2^\ell, \boldsymbol{\Sigma}_3^\ell\right),
\tag{8}
$$

where $\boldsymbol{\Sigma}_1^\ell \in \mathbb{R}^{D_1^\ell \times D_1^\ell}$, $\boldsymbol{\Sigma}_2^\ell \in \mathbb{R}^{D_2^\ell \times D_2^\ell}$, and $\boldsymbol{\Sigma}_3^\ell \in \mathbb{R}^{T \times T}$ are the mode-1, mode-2, and mode-3 covariance matrices, respectively. Specifically, in the tensor normal prior, the row covariance matrix $\boldsymbol{\Sigma}_1^\ell$ models the relationships between features (feature covariance), the column covariance matrix $\boldsymbol{\Sigma}_2^\ell$ models the relationships between classes (class covariance), and the mode-3 covariance matrix $\boldsymbol{\Sigma}_3^\ell$ models the relationships between tasks in the $\ell$-th layer network parameters $\{\mathbf{W}^{1,\ell}, \ldots, \mathbf{W}^{T,\ell}\}$. A common strategy used by previous methods is to use identity covariance for feature covariance [31, 8] and class covariance [2], which implicitly assumes independent features and classes and cannot capture the dependencies between them. This work learns all feature covariance, class covariance, task covariance and all network parameters from data to build robust multilinear task relationships.

We integrate the CNN error functional (6) and tensor normal prior (8) into MAP estimation (7) and taking negative logarithm, which leads to the MAP estimation of the network parameters $\mathcal{W}$, a regularized optimization problem for Multilinear Relationship Network (MRN) formally writing as

$$
\begin{aligned}
\min_{f_t |_{t=1}^T, \boldsymbol{\Sigma}_k^\ell |_{k=1}^K} &\sum_{t=1}^T \sum_{n=1}^{N_t} J\left(f_t\left(\mathbf{x}_n^t\right), \mathbf{y}_n^t\right) \\
&+ \frac{1}{2} \sum_{\ell \in \mathcal{L}} \left(\mathrm{vec}(\mathcal{W}^\ell)^{\mathsf{T}} (\boldsymbol{\Sigma}_{1:K}^\ell)^{-1} \mathrm{vec}(\mathcal{W}^\ell) - \sum_{k=1}^K \frac{D^\ell}{D_k^\ell} \ln\left(|\boldsymbol{\Sigma}_k^\ell|\right)\right),
\end{aligned}
\tag{9}
$$

where $D^\ell = \prod_{k=1}^K D_k^\ell$ and $K = 3$ is the number of modes in parameter tensor $\mathcal{W}$, which could be $K = 4$ for the convolutional layers (width, height, number of feature maps, and number of tasks); $\boldsymbol{\Sigma}_{1:3}^\ell = \boldsymbol{\Sigma}_1^\ell \otimes \boldsymbol{\Sigma}_2^\ell \otimes \boldsymbol{\Sigma}_3^\ell$ is the Kronecker product of the feature covariance $\boldsymbol{\Sigma}_1^\ell$, class covariance $\boldsymbol{\Sigma}_2^\ell$, and task covariance $\boldsymbol{\Sigma}_3^\ell$. Moreover, we can assume shared task relationship across different layers as $\boldsymbol{\Sigma}_3^\ell = \boldsymbol{\Sigma}_3$, which enhances connection between task relationships on features $fc7$ and classifiers $fc8$.

## 4.2 Algorithm

The optimization problem (9) is jointly non-convex with respect to the parameter tensors $\mathcal{W}$ as well as feature covariance $\boldsymbol{\Sigma}_1^\ell$, class covariance $\boldsymbol{\Sigma}_2^\ell$, and task covariance $\boldsymbol{\Sigma}_3^\ell$. Thus, we alternatively optimize

one set of variables with the others fixed. We first update $\mathbf{W}^{t,\ell}$, the parameter of task-$t$ in layer-$\ell$. When training deep CNN by back-propagation, we only require the gradient of the objective function (denoted by $O$) in Equation (10) w.r.t. $\mathbf{W}^{t,\ell}$ on each data point $(\mathbf{x}_n^t, \mathbf{y}_n^t)$, which can be computed as

$$\frac{\partial O\left(\mathbf{x}_n^t, \mathbf{y}_n^t\right)}{\partial \mathbf{W}^{t,\ell}} = \frac{\partial J\left(f_t\left(\mathbf{x}_n^t\right), \mathbf{y}_n^t\right)}{\partial \mathbf{W}^{t,\ell}} + \left[(\mathbf{\Sigma}_{1:3}^\ell)^{-1}\mathrm{vec}\left(\mathcal{W}^\ell\right)\right]_{..t}, \tag{10}$$

where $[(\mathbf{\Sigma}_{1:3}^\ell)^{-1}\mathrm{vec}\left(\mathcal{W}^\ell\right)]_{..t}$ is the $(:,:,t)$ slice of a tensor folded from elements $(\mathbf{\Sigma}_{1:3}^\ell)^{-1}\mathrm{vec}(\mathcal{W}^\ell)$ that are corresponding to parameter matrix $\mathbf{W}^{t,\ell}$. Since training a deep CNN requires a large amount of labeled data, which is prohibitive for many multi-task learning problems, we fine-tune from an AlexNet model pre-trained on ImageNet as in [28]. In each epoch, after updating $\mathcal{W}$, we can update the feature covariance $\mathbf{\Sigma}_1^\ell$, class covariance $\mathbf{\Sigma}_2^\ell$, and task covariance $\mathbf{\Sigma}_3^\ell$ by the flip-flop algorithm as

$$\mathbf{\Sigma}_1^\ell = \frac{1}{D_2^\ell T}(\mathcal{W}^\ell)_{(1)}\left(\mathbf{\Sigma}_3^\ell \otimes \mathbf{\Sigma}_2^\ell\right)^{-1}(\mathcal{W}^\ell)_{(1)}^\mathsf{T} + \epsilon\mathbf{I}_{D_1^\ell},$$

$$\mathbf{\Sigma}_2^\ell = \frac{1}{D_1^\ell T}(\mathcal{W}^\ell)_{(2)}\left(\mathbf{\Sigma}_3^\ell \otimes \mathbf{\Sigma}_1^\ell\right)^{-1}(\mathcal{W}_\ell)_{(2)}^\mathsf{T} + \epsilon\mathbf{I}_{D_2^\ell}, \tag{11}$$

$$\mathbf{\Sigma}_3^\ell = \frac{1}{D_1^\ell D_2^\ell}(\mathcal{W}^\ell)_{(3)}\left(\mathbf{\Sigma}_2^\ell \otimes \mathbf{\Sigma}_1^\ell\right)^{-1}(\mathcal{W}^\ell)_{(3)}^\mathsf{T} + \epsilon\mathbf{I}_T.$$

where the last term of each update equation is a small penalty traded off by $\epsilon$ for numerical stability.

However, the above updating equations (11) are computationally prohibitive, due to the dimension explosion of the Kronecker product, e.g. $\mathbf{\Sigma}_2^\ell \otimes \mathbf{\Sigma}_1^\ell$ is of dimension $D_1^\ell D_2^\ell \times D_1^\ell D_2^\ell$. To speed up computation, we will use the following rules of Kronecker product: $(\mathbf{A} \otimes \mathbf{B})^{-1} = \mathbf{A}^{-1} \otimes \mathbf{B}^{-1}$ and $\left(\mathbf{B}^\mathsf{T} \otimes \mathbf{A}\right)\mathrm{vec}\left(\mathbf{X}\right) = \mathrm{vec}\left(\mathbf{A}\mathbf{X}\mathbf{B}\right)$. Taking the computation of $\mathbf{\Sigma}_3^\ell \in \mathbb{R}^{T \times T}$ as an example, we have

$$\begin{aligned}(\Sigma_3^\ell)_{ij} &= \frac{1}{D_1^\ell D_2^\ell}(\mathcal{W}^\ell)_{(3),i\cdot}\left(\mathbf{\Sigma}_2^\ell \otimes \mathbf{\Sigma}_1^\ell\right)^{-1}(\mathcal{W}^\ell)_{(3),j\cdot}^\mathsf{T} + \epsilon I_{ij} \\ &= \frac{1}{D_1^\ell D_2^\ell}(\mathcal{W}^\ell)_{(3),i\cdot}\mathrm{vec}\left((\mathbf{\Sigma}_1^\ell)^{-1}\mathcal{W}_{\cdot\cdot j}^\ell(\mathbf{\Sigma}_2^\ell)^{-1}\right) + \epsilon I_{ij},\end{aligned} \tag{12}$$

where $(\mathcal{W}^\ell)_{(3),i\cdot}$ denotes the $i$-th row of the mode-3 matricization of tensor $\mathcal{W}^\ell$, and $\mathcal{W}_{\cdot\cdot j}^\ell$ denotes the $(:,:,j)$ slice of tensor $\mathcal{W}^\ell$. We can derive that updating $\mathbf{\Sigma}_3^\ell$ has a computational complexity of $O\left(T^2 D_1^\ell D_2^\ell \left(D_1^\ell + D_2^\ell\right)\right)$, similarly for $\mathbf{\Sigma}_1^\ell$ and $\mathbf{\Sigma}_2^\ell$. The total computational complexity of updating covariance matrices $\mathbf{\Sigma}_k^\ell|_{k=1}^3$ will be $O\left(D_1^\ell D_2^\ell T \left(D_1^\ell D_2^\ell + D_1^\ell T + D_2^\ell T\right)\right)$, which is still expensive.

A key to computation speedup is that the covariance matrices $\mathbf{\Sigma}_k^\ell|_{k=1}^3$ should be low-rank, since the features and tasks are enforced to be correlated for multi-task learning. Thus, the inverses of $\mathbf{\Sigma}_k^\ell|_{k=1}^3$ do not exist in general and we have to compute the generalized inverses using eigendecomposition. We perform eigendecomposition for each $\mathbf{\Sigma}_k^\ell$ and maintain all eigenvectors with eigenvalues greater than zero. The rank $r$ of the eigen-reconstructed covariance matrices should be $r \leq \min(D_1^\ell, D_2^\ell, T)$. Thus, the total computational complexity for $\mathbf{\Sigma}_k^\ell|_{k=1}^3$ is reduced to $O\left(rD_1^\ell D_2^\ell T \left(D_1^\ell + D_2^\ell + T\right)\right)$. It is straight-forward to see the computational complexity of updating the parameter tensor $\mathcal{W}$ is the cost of back-propagation in standard CNNs plus the cost for computing the gradient of regularization term by Equation (10), which is $O\left(rD_1^\ell D_2^\ell T \left(D_1^\ell + D_2^\ell + T\right)\right)$ given generalized inverses $(\mathbf{\Sigma}_k^\ell)^{-1}|_{k=1}^3$.

### 4.3 Discussion

The proposed Multilinear Relationship Network (MRN) is very flexible and can be easily configured to deal with different network architectures and multi-task learning scenarios. For example, replacing the network backbone from AlexNet to VGGnet [24] boils down to configuring task-specific layers $\mathcal{L} = \{fc7, fc8\}$, where $fc7$ is the last feature layer while $fc8$ is the classifier layer in the VGGnet. The architecture of MRN in Figure 1 can readily cope with homogeneous multi-task learning where all tasks share the same output space. It can cope with heterogeneous multi-task learning where different tasks have different output spaces by setting $\mathcal{L} = \{fc7\}$, by only considering feature layers.

The multilinear relationship learning in Equation (9) is a general framework that readily subsumes many classical multi-task learning methods as special cases. Many regularized multi-task algorithms can be classified into two main categories: learning with feature covariances [1, 2, 6, 5] and learning

with task relations [10, 14, 29, 31, 15, 17, 8]. Learning with feature covariances can be viewed as a representative formulation in feature-based methods while learning with task relations is for parameter-based methods [30]. More specifically, previous multi-task feature learning methods [1, 2] can be viewed as a special case of Equation (9) by setting all covariance matrices but the feature covariance to identity matrix, i.e. $\Sigma_k = \mathbf{I}|_{k=2}^{K}$; and previous multi-task relationship learning methods [31, 8] can be viewed as a special case of Equation (9) by setting all covariance matrices but the task covariance to identity matrix, i.e. $\Sigma_k = \mathbf{I}|_{k=1}^{K-1}$. The proposed MRN is more general in the architecture perspective in dealing with parameter tensors in multiple layers of deep neural networks.

It is noteworthy to highlight a concurrent work on multi-task deep learning using tensor decomposition [27], which is feature-based method that explicitly learns the low-rank shared parameter subspace. The proposed multilinear relationship across parameter tensors can be viewed as a strong alternative to the tensor decomposition, with the advantage to explicitly model the positive and negative relations across features and tasks. As a defense of [27], the tensor decomposition can extract finer-grained feature relations (what to share and how much to share) than the proposed multilinear relationships.

# 5 Experiments

We compare MRN with state-of-the-art multi-task and deep learning methods to verify the efficacy of learning transferable features and multilinear task relationships. Codes and datasets will be released.

## 5.1 Setup

**Office-Caltech** [12]   This dataset is the standard benchmark for multi-task learning and transfer learning. The Office part consists of 4,652 images in 31 categories collected from three distinct domains (tasks): *Amazon* (**A**), which contains images downloaded from amazon.com, *Webcam* (**W**) and *DSLR* (**D**), which are images taken by Web camera and digital SLR camera under different environmental variations. This dataset is organized by selecting the 10 common categories shared by the Office dataset and the Caltech-256 (**C**) dataset [12], hence it yields four multi-class learning tasks.

**Office-Home**[1] [26]   This dataset is to evaluate transfer learning algorithms using deep learning. It consists of images from 4 different domains: Artistic images (**A**), Clip Art (**C**), Product images (**P**) and Real-World images (**R**). For each domain, the dataset contains images of 65 object categories collected in office and home settings.

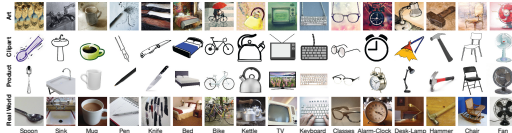

Figure 2: Examples of the Office-Home dataset.

**ImageCLEF-DA**[2]   This dataset is the benchmark for ImageCLEF domain adaptation challenge, organized by selecting the 12 common categories shared by the following four public datasets (tasks): Caltech-256 (**C**), ImageNet ILSVRC 2012 (**I**), Pascal VOC 2012 (**P**), and Bing (**B**). All three datasets are evaluated using DeCAF$_7$ [9] features for shallow methods and original images for deep methods.

We compare MRN with standard and state-of-the-art methods: Single-Task Learning (**STL**), Multi-Task Feature Learning (**MTFL**) [2], Multi-Task Relationship Learning (**MTRL**) [31], Robust Multi-Task Learning (**RMTL**) [5], and Deep Multi-Task Learning with Tensor Factorization (**DMTL-TF**) [27]. STL performs per-task classification in separate deep networks without knowledge transfer. MTFL extracts the low-rank shared feature representations by learning feature covariance. RMTL extends MTFL to further capture the task relationships using a low-rank structure and identify outlier tasks using a group-sparse structure. MTRL captures the task relationships using task covariance of a matrix normal distribution. DMTL-TF tackles multi-task deep learning by tensor factorization, which learns shared feature subspace instead of multilinear task relationship in multilayer parameter tensors.

To go deep into the efficacy of jointly learning transferable features and multilinear task relationships, we evaluate two MRN variants: (1) **MRN**$_8$, MRN using only one network layer $fc8$ for multilinear relationship learning; **(2) MRN**$_t$, MRN using only task covariance $\Sigma_3$ for single-relationship learning. The proposed MRN model can natively deal with multi-class problems using the parameter tensors. However, most shallow multi-task learning methods such as MTFL, RMTL and MTRL are formulated

Table 1: Classification accuracy on *Office-Caltech* with standard evaluation protocol (AlexNet).

| Method | 5% | | | | | 10% | | | | | 20% | | | | |
|---|---|---|---|---|---|---|---|---|---|---|---|---|---|---|---|
| | A | W | D | C | Avg | A | W | D | C | Avg | A | W | D | C | Avg |
| STL (AlexNet) | 88.9 | 73.0 | 80.4 | 88.7 | 82.8 | 92.2 | 80.9 | 88.2 | 88.9 | 87.6 | 91.3 | 83.3 | 93.7 | **94.9** | 90.8 |
| MTFL [2] | 90.0 | 78.9 | 90.2 | 86.9 | 86.5 | 92.4 | 85.3 | 89.5 | **89.2** | 89.1 | 93.5 | 89.0 | 95.2 | 92.6 | 92.6 |
| RMTL [6] | 91.3 | 82.3 | 88.8 | **89.1** | 87.9 | 92.6 | 85.2 | 93.3 | 87.2 | 89.6 | 94.3 | 87.0 | 96.7 | 93.4 | 92.4 |
| MTRL [31] | 86.4 | 83.0 | 95.1 | **89.1** | 88.4 | 91.1 | 87.1 | 97.0 | 87.6 | 90.7 | 90.0 | 88.8 | 99.2 | 94.3 | 93.1 |
| DMTL-TF [27] | 91.2 | 88.3 | 92.5 | 85.6 | 89.4 | 92.2 | 91.9 | 97.4 | 86.8 | 92.0 | 92.6 | 97.6 | 94.5 | 88.4 | 93.3 |
| **MRN$_8$** | 91.7 | 96.4 | 96.9 | 86.5 | 92.9 | 92.7 | 97.1 | 97.3 | 86.6 | 93.4 | 93.2 | 96.9 | 99.4 | 82.8 | 94.4 |
| **MRN$_t$** | 91.1 | 96.3 | 97.4 | 86.1 | 92.7 | 92.5 | 97.7 | 96.6 | 86.7 | 93.4 | 91.9 | 96.6 | 95.9 | 90.0 | 93.6 |
| **MRN (full)** | **92.5** | **97.5** | **97.9** | 87.5 | **93.8** | **93.6** | **98.6** | **98.6** | 87.3 | **94.5** | **94.4** | **98.3** | **99.9** | 89.1 | **95.5** |

Table 2: Classification accuracy on *Office-Home* with standard evaluation protocol (VGGnet).

| Method | 5% | | | | | 10% | | | | | 20% | | | | |
|---|---|---|---|---|---|---|---|---|---|---|---|---|---|---|---|
| | A | C | P | R | Avg | A | C | P | R | Avg | A | C | P | R | Avg |
| STL (VGGnet) | 35.8 | 31.2 | 67.8 | 62.5 | 49.3 | 51.0 | 40.7 | 75.0 | 68.8 | 58.9 | 56.1 | 54.6 | 80.4 | 71.8 | 65.7 |
| MTFL [2] | 40.1 | 30.4 | 61.5 | 59.5 | 47.9 | 50.3 | 35.0 | 66.3 | 65.0 | 54.2 | 55.2 | 38.8 | 69.1 | 70.0 | 58.3 |
| RMTL [6] | 42.3 | 32.8 | 62.3 | 60.6 | 49.5 | 49.7 | 34.6 | 65.9 | 64.6 | 53.7 | 55.2 | 39.2 | 69.6 | 70.5 | 58.6 |
| MTRL [31] | 42.7 | 33.3 | 62.9 | 61.3 | 50.1 | 51.6 | 36.3 | 67.7 | 66.3 | 55.5 | 55.8 | 39.9 | 70.2 | 71.2 | 59.3 |
| DMTL-TF [27] | 49.2 | 34.5 | 67.1 | 62.9 | 53.4 | 57.2 | 42.3 | 73.6 | 69.9 | 60.8 | 58.3 | **56.1** | 79.3 | 72.1 | 66.5 |
| **MRN$_8$** | 52.7 | 34.7 | 70.1 | 67.6 | 56.3 | 59.1 | 42.7 | 75.1 | 72.8 | 62.4 | 58.4 | 55.6 | 80.4 | 72.4 | 66.7 |
| **MRN$_t$** | 52.0 | 34.0 | 69.9 | 66.8 | 55.7 | 58.6 | 42.6 | 74.9 | 72.4 | 62.1 | 57.7 | 54.8 | 80.2 | 71.6 | 66.1 |
| **MRN (full)** | **53.3** | **36.4** | **70.5** | **67.7** | **57.0** | **59.9** | **42.7** | **76.3** | **73.0** | **63.0** | **58.5** | 55.6 | **80.7** | **72.8** | **66.9** |

only for binary-class problems, due to the difficulty in dealing with order-3 parameter tensors for multi-class problems. We adopt one-vs-rest strategy to enable them working on multi-class datasets.

We follow the standard evaluation protocol [31, 5] for multi-task learning and randomly select 5%, 10%, and 20% samples from each task as training set and use the rest of the samples as test set. We compare the average classification accuracy for all tasks based on five random experiments, where standard errors are generally less than $\pm0.5\%$, which are not significant and thus are not reported for space limitation. We conduct model selection for all methods using five-fold cross-validation on the training set. For deep learning methods, we adopt AlexNet [16] and VGGnet [24], fix convolutional layers $conv1$–$conv5$, fine-tune fully-connected layers $fc6$–$fc7$, and train classifier layer $fc8$ via back-propagation. As the classifier layer is trained from scratch, we set its learning rate to be 10 times that of the other layers. We use mini-batch stochastic gradient descent (SGD) with 0.9 momentum and learning rate decaying strategy, and select learning rate between $10^{-5}$ and $10^{-2}$ by stepsize $10^{\frac{1}{2}}$.

## 5.2 Results

The multi-task classification results on the Office-Caltech, Office-Home and ImageCLEF-DA datasets based on 5%, 10%, and 20% sampled training data are shown in Tables 1, 2 and 3, respectively. We observe that the proposed MRN model significantly outperforms the comparison methods on most multi-task problems. The substantial accuracy improvement validates that our multilinear relationship networks through multilayer and multilinear relationship learning is able to learn both transferable features and adaptive task relationships, which enables effective and robust multi-task deep learning.

We can make the following observations from the results. **(1)** Shallow multi-task learning methods MTFL, RMTL, and MTRL outperform single-task deep learning method STL in most cases, which confirms the efficacy of learning multiple tasks by exploiting shared structures. Among the shallow multi-task methods, MTRL gives the best accuracies, showing that exploiting task relationship may be more effective than extracting shared feature subspace for multi-task learning. It is worth noting that, although STL cannot learn from knowledge transfer, it can be fine-tuned on each task to improve performance, and thus when the number of training samples are large enough and when different tasks are dissimilar enough (e.g. Office-Home dataset), STL may outperform shallow multi-task learning methods, as evidenced by the results in Table 2. **(2)** Deep multi-task learning method DMTL-TF outperforms shallow multi-task learning methods with deep features as input, which confirms the importance of learning deep transferable features to enable knowledge transfer across tasks. However, DMTL-TF only learns the shared feature subspace based on tensor factorization of the network parameters, while the task relationships in multiple network layers are not captured. This may result in negative-transfer in the feature layers [28] and under-transfer in the classifier layers. Negative-transfer can be witnessed by comparing multi-task methods with single-task methods: if multi-task learning methods yield lower accuracy in some of the tasks, then negative-transfer arises.

Table 3: Classification accuracy on *ImageCLEF-DA* with standard evaluation protocol (AlexNet).

| Method | 5% | | | | | 10% | | | | | 20% | | | | |
|---|---|---|---|---|---|---|---|---|---|---|---|---|---|---|---|
| | C | I | P | B | Avg | C | I | P | B | Avg | C | I | P | B | Avg |
| STL (AlexNet) | 77.4 | 60.3 | 48.0 | 45.0 | 57.7 | 78.9 | 70.5 | 48.1 | 41.8 | 59.8 | 83.3 | 74.9 | 49.2 | 47.1 | 63.6 |
| MTFL [2] | 79.9 | 68.6 | 43.4 | 41.5 | 58.3 | 82.9 | 71.4 | 56.7 | 41.7 | 63.2 | 83.1 | 72.2 | 54.5 | 52.5 | 65.6 |
| RMTL [6] | 81.1 | 71.3 | 52.4 | 40.9 | 61.4 | 81.5 | 71.7 | 55.6 | 45.3 | 63.5 | 83.3 | 73.3 | 53.7 | 49.2 | 64.9 |
| MTRL [31] | 80.8 | 68.4 | 51.9 | 42.9 | 61.0 | 83.1 | 72.7 | 54.5 | 45.5 | 63.9 | 83.7 | 75.5 | 57.5 | 49.4 | 66.5 |
| DMTL-TF [27] | 87.9 | 70.0 | 58.1 | 34.1 | 62.5 | **89.1** | 82.1 | 58.7 | 48.0 | 69.5 | 91.7 | 80.0 | 63.2 | 54.1 | 72.2 |
| **MRN$_8$** | 87.0 | 74.4 | 61.8 | 47.6 | 67.7 | **89.1** | 82.2 | 64.4 | 49.3 | 71.2 | 91.1 | **84.1** | 65.7 | 54.1 | 73.7 |
| **MRN$_t$** | 88.5 | 73.5 | 63.3 | **51.1** | 69.1 | 88.0 | 83.1 | 67.4 | 54.8 | 73.3 | 91.1 | 83.5 | 65.7 | 55.7 | 74.0 |
| **MRN (full)** | **89.6** | **76.9** | **65.4** | 49.4 | **70.3** | 88.1 | **84.6** | **68.7** | **55.6** | **74.3** | **92.8** | 83.3 | **67.4** | **57.8** | **75.3** |

We go deeper into MRN by reporting the results of the two MRN variants: MRN$_8$ and MRN$_t$, all significantly outperform the comparison methods but generally underperform MRN (full), which verify our motivation that jointly learning transferable features and multilinear task relationships can bridge multiple tasks more effectively. **(1)** The disadvantage of MRN$_8$ is that it does not learn the task relationship in the lower layers $fc7$, which are not safely transferable and may result in negative transfer [28]. **(2)** The shortcoming of MRN$_t$ is that it does not learn the multilinear relationship of features, classes and tasks, hence the learned relationships may only capture the task covariance without capturing the feature covariance and class covariance, which may lose some intrinsic relations.

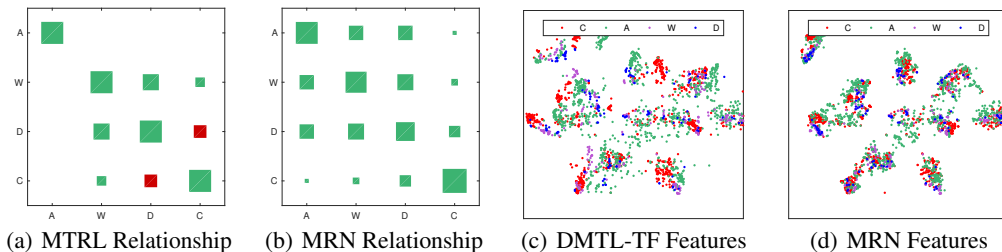

(a) MTRL Relationship   (b) MRN Relationship   (c) DMTL-TF Features   (d) MRN Features

Figure 3: Hinton diagram of task relationships (a)(b) and t-SNE embedding of deep features (c)(d).

## 5.3 Visualization Analysis

We show that MRN can learn more reasonable task relationships with deep features than MTRL with shallow features, by visualizing the Hinton diagrams of task covariances learned by MTRL and MRN ($\Sigma_3^{fc8}$) in Figures 3(a) and 3(b), respectively. Prior knowledge on task similarity in the Office-Caltech dataset [12] describes that tasks **A**, **W** and **D** are more similar with each other while they are relatively dissimilar to task **C**. MRN successfully captures this prior task relationship and enhances the task correlation across dissimilar tasks, which enables stronger transferability for multi-task learning. Furthermore, all tasks are positively correlated (green color) in MRN, implying that all tasks can better reinforce each other. However, some of the tasks (**D** and **C**) are still negatively correlated (red color) in MTRL, implying these tasks should be drawn far apart and cannot improve with each other.

We illustrate the feature transferability by visualizing in Figures 3(c) and 3(d) the t-SNE embeddings [18] of the images in the Office-Caltech dataset with DMTL-TF features and MRN features, respectively. Compared with DMTL-TF features, the data points with MRN features are discriminated better across different categories, i.e., each category has small intra-class variance and large inter-class margin; the data points are also aligned better across different tasks, i.e. the embeddings of different tasks overlap well, implying that different tasks reinforce each other effectively. This verifies that with multilinear relationship learning, MRN can learn more transferable features for multi-task learning.

## 6 Conclusion

This paper presented multilinear relationship networks (MRN) that integrate deep neural networks with tensor normal priors over the network parameters of all task-specific layers, which model the task relatedness through the covariance structures over tasks, classes and features to enable transfer across related tasks. An effective learning algorithm was devised to jointly learn transferable features and multilinear relationships. Experiments testify that MRN yields superior results on standard datasets.

## Acknowledgments

This work was supported by the National Key R&D Program of China (2016YFB1000701), National Natural Science Foundation of China (61772299, 61325008, 61502265, 61672313) and TNList Fund.

## Footnotes

[1]http://hemanthdv.org/OfficeHome-Dataset

[2]http://imageclef.org/2014/adaptation

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
