[Reviews · NeurIPS 2017]

Reviewer 1



I found the paper rather interesting. The idea of using tensor-variate priors for the weights was excellent. However, it could have been motivated better. In particular, the comparison with matrix-variate priors should have been explained better and the discussion on ll.237-242 on previous work should be expanded a lot more, possibly in the related works section. The experimental section seems ok, however I would strongly suggest to remove results on the Office-Caltech dataset as the dataset has served its purpose and is not useful anymore for most deep models (very few datasamples, label pollution etc).

Reviewer 2



This paper proposes a solution to a very interesting problem, namely multi-task learning. The authors tackles learning task similarities. It is known to be very hard problem and it has been ignored in many problems that were proposed to solve multi-task learning. The authors proposes a Bayesian method by using tensor normal priors. The paper is well written and well connected to literature. The idea is quite novel. The authors supply a set of convincing experiments and to support their method. Although I am quite positive about paper, I would like to get information to the following questions: 1) Although experiments are convincing, I would like to see the standard deviations (stds)for experiments? Since the dataset sizes are small then reporting stds are quite important. 2) The authors uses AlexNet as a baseline. However, AlexNet architecture is not anymore the state of the art architecture. It would be great if authors uses GoogleNet or something similar and show the improvement of their algorithm.

Reviewer 3



The paper proposes an interesting approach for improving multi-task learning by modeling task relationships. The paper is well-written, compares to relevant benchmarks on relevant datasets, and provides additional insight about what the classifier is learning. While the results are impressive in the settings explored, the method seems unlikely to scale well to a large number of tasks or lifelong settings.